# Web-Based Reporting of Post-Vaccination Symptoms for Inactivated COVID-19 Vaccines in Jordan: A Cross-Sectional Study

**DOI:** 10.3390/vaccines11010044

**Published:** 2022-12-25

**Authors:** Razan I. Nassar, Muna Barakat, Samar Thiab, Feras El-Hajji, Hiba Barqawi, Waseem El-Huneidi, Iman A. Basheti, Eman Abu-Gharbieh

**Affiliations:** 1Department of Clinical Pharmacy and Therapeutics, Faculty of Pharmacy, Applied Science Private University, Amman 11931, Jordan; 2Department of Pharmaceutical Chemistry and Pharmacognosy, Faculty of Pharmacy, Applied Science Private University, Amman 11931, Jordan; 3Sharjah Institute for Medical Research, University of Sharjah, Sharjah 27272, United Arab Emirates; 4Department of Clinical Sciences, College of Medicine, University of Sharjah, Sharjah 27272, United Arab Emirates; 5Department of Basic Medical Sciences, College of Medicine, University of Sharjah, Sharjah 27272, United Arab Emirates; 6Department of Clinical Pharmacy and Therapeutics, Faculty of Pharmacy, The University of Sydney, Sydney, NSW 2006, Australia

**Keywords:** coronavirus, COVID-19, side effects, vaccines, inactivated COVID-19 vaccine, Jordan

## Abstract

Background: The perception of COVID-19 vaccines as being unsafe is a major barrier to receiving the vaccine. Providing the public with accurate data regarding the vaccines would reduce vaccine hesitancy. Methods: A cross-sectional study was conducted to collect data on the side effects experienced by the vaccinated population to assess the safety of the inactivated COVID-19 vaccine. Results: The majority of the study participants (n = 386) were female (71.9%), and 38.6% of them were under 30 years old. Around half of the participants (52.8%) reported side effects after receiving the inactivated COVID-19 vaccine. Fatigue (85.1%), a sore arm at the site of the injection (82.1%), and discomfort (67.2%) were the most commonly reported side effects after the first dose. Reporting side effects was significantly associated with the female sex (*p*-value = 0.027). Significant associations between being female and experiencing chills, muscle or joint pain, anorexia, drowsiness, and hair loss were also found, as well as being above the age of 30 and experiencing a cough. Being a smoker was significantly associated with experiencing a cough, and a headache. Furthermore, chills, and a sore throat were significantly associated with individuals who had not been infected before. Conclusion: Mild side effects were reported after receiving the inactivated COVID-19 vaccine. Fatigue was the most commonly reported side effect. Females, older adults, smokers, and those who had never been infected with COVID-19 had a greater susceptibility to certain side effects.

## 1. Introduction

When a disease is declared to be a pandemic, healthcare providers, clinical researchers, and pharmaceutical manufacturers rush to find a cure or a strategy of prevention to reduce the spread of the disease and its related death toll. This includes the development of vaccines to control the spread of the disease. The World Health Organization (WHO) declared the outbreak of coronavirus disease 2019 (COVID-19) to be a pandemic in March 2020 [1]. By July 2021, more than ten COVID-19 vaccines were approved by at least one regulatory authority for emergency use [2,3,4,5]. These vaccines were developed to reduce virus transmission and provide active acquired immunity. Financial support from private consortia and the governments of some countries allowed the rapid development of COVID-19 vaccines compared to traditional vaccines [4,6]. The COVID-19 vaccines are categorised into different types: whole attenuated or inactivated virus ones, viral vector, protein-based, and nucleic acid ones [2,6,7].

Around 70% of the world’s population has received at least one dose of a COVID-19 vaccine, and in Jordan, around 50% of the population is vaccinated [8]. Among the vaccines that are available in Jordan is the inactivated COVID-19 vaccine. These vaccines are developed by exposing the virus to chemicals such as formaldehyde, heat, or radiation to make it stable and non-infectious so that it does not proliferate in vivo, but it still triggers an immune response [2,4,7]. An advantage to these vaccines is that they do not require cold chains for distribution because they can be freeze-dried [4]. The technology behind developing inactivated vaccines has led to numerous vaccines in the past, including seasonal influenza vaccines [2]. However, the main disadvantage of this type of vaccine is its short immune memory duration, thus, there is a need to inoculate with higher amounts of the vaccine or associate it with an adjuvant to boost the acquired immunity [2,6].

According to the Strategic Advisory Group of Experts (SAGE), vaccine hesitancy is a ‘delay in accepting or refusing vaccination, in spite of the availability of immunisation services’ [9]. After conducting several studies to assess vaccine hesitancy, it was revealed that it is a common worldwide phenomenon [10,11]. Vaccine intake results from five factors (5As): accessibility, affordability, awareness, acceptance, and activation [12]. Limited knowledge, a lack of awareness, and the perceived risk versus benefits are some of the most commonly reported reasons for vaccine hesitancy [13,14]; these reasons can apply to COVID-19 vaccine hesitation, as evidenced by recent studies that reveal a strong correlation between the intent to receive the vaccine and it’s perceived safety [15]. Furthermore, the rapid development of the COVID-19 vaccines has been identified as a major concern, thus, it is a barrier to receiving the vaccine and the side effects associated with the vaccine [16]. 

Globally, there was a need to increase public confidence in COVID-19 vaccines. Accordingly, the Jordanian government took many actions to increase the trust in the COVID-19 vaccine, including a countrywide immunisation campaign coordinated by the Ministry of Health and the National Center for Security and Crisis Management [16,17]. Other measures that can be taken to increase public trust in vaccines include providing accurate data regarding the COVID-19 vaccine by establishing a transparent database to report the post-vaccination side effects [16].

People’s attitudes towards vaccination have changed over the era of the COVID-19 pandemic. No specific study has looked closely into this change of attitude, however, an indirect evaluation of the reported beliefs can provide such an insight. A cross-sectional study (August 2020) was conducted to evaluate the perception of people in Jordan regarding the COVID-19 vaccines and assess their hesitancy toward receiving the COVID-19 vaccine. Out of the participants (n = 1287), more than half of them (n = 665) reported not having adequate information about the COVID-19 vaccine benefits. Moreover, 64% of them preferred to achieve natural immunity [18]. In December 2020, a cross-sectional study was conducted to assess the attitudes toward COVID-19 vaccines among several countries, and the acceptance rate for the COVID-19 vaccine in Jordan was found to be 28.4% [19]. One explanation for the low COVID-19 vaccine acceptance could be conspiracy beliefs which appear to worsen over time. A cross-sectional study conducted in April 2020 revealed that 47.9% of the participants (n = 3150) think the COVID-19 pandemic is a part of a global conspiracy theory [20]. After eight months, in December 2020, another cross-sectional study was conducted, and the percentage increased, as 58.5% of the participants believed that COVID-19 is a man-made disease [19]. Such a discrepancy in COVID-19 vaccine acceptance might be attributed to the participants’ age, sex, background, and educational level, for example, the study findings of a cross-sectional study conducted to assess attitudes and perceptions of Jordanian healthcare providers to the COVID-19 vaccine showed that the acceptance level ranges from 42.6% for nurses to 83.3% for physicians [21]. 

This study aimed to assess the safety of the inactivated COVID-19 vaccine and reveal the association between certain side effects and different parameters, which in turn would provide accurate data regarding the COVID-19 vaccine and help the public understand what to expect after receiving the vaccine. This is the first study conducted among the Jordanian population to assess the side effects specifically associated with the inactivated COVID-19 vaccine. Other published studies in Jordan assessed different types of COVID-19 vaccines, with none of them focusing on the inactivated COVID-19 vaccine. 

## 2. Methods

### 2.1. Study Design and Participants

A cross-sectional study was carried out in August 2021 to collect data regarding the side effects of the inactivated COVID-19 vaccine (Sinopharm/Beijing^®^, Beijing, China) among the Jordanian population. The survey was developed and disseminated digitally using *Google Forms* (Google^®^, Menlo Park, CA, USA) to Jordanian inhabitants who had taken at least the first dose of the inactivated COVID-19 vaccine who were deemed eligible to participate in the study.

The participants received no financial compensation to reduce the self-selection bias. The importance of participating in this research, in educating the population and reducing hesitancy toward vaccines, was clearly explained. Furthermore, participation in the study was voluntary. The participants provided informed consent to participate in the research. 

Ethical approval for this study was obtained from the Scientific and Ethics Committee of the Faculty of Pharmacy, Applied Science Private University (Approval Number: 2021-PHA-31). 

### 2.2. Survey Development

Following an extensive review of the literature, a broad spectrum of potential side effects following the administration of the inactivated COVID-19 vaccine was identified, thus, a pool of questions was generated from various sources to assist in constructing the first draft of the survey (1–3). To meet the study objectives, the research team developed the survey based on the available information regarding the side effects of the inactivated COVID-19 vaccine.

To ensure face and content validity, the first draft of the survey was validated by an expert panel who evaluated the questions’ comprehension, relevancy, and word clarity. The expert panel included five independent academics who were randomly selected from a list of 30 academics and who worked at different higher education institutions. The academics were selected based on their experience (minimum 10 years) in related research areas. In addition to the five academics, two specialist pulmonologists were requested to validate the survey. The experts were invited to assess the suitability of the words, appropriateness of the content, consistency of the layout and style, relevancy of the survey items to the study objectives, and whether the items were related to the study’s aim. Furthermore, they confirmed that the survey was free from medical jargon and complicated terminology. The amendments were conducted based on their feedback. A pilot for the survey was conducted, and necessary refinements were made. The survey questions were revised as a final point in the survey development and to make the study appropriate for online administration.

The survey’s final edition consisted of two primary sections addressing the aspects of interest. Section 1 included items to collect information regarding the participants’ socio-demographics (sex, age, marital status, living place, nationality, educational level, employment, and smoking status). In this section, the participants were also questioned if they had previously been infected with COVID-19. Section 2 included items to collect data regarding the side effects of being vaccinated with the inactivated COVID-19 vaccine. The following data were collected: the month in which the vaccine was taken (first dose), whether the participant was vaccinated with one or two dose(s), the side effects that the participants experienced after vaccination, and whether the participants experienced any severe side effects which required their hospitalisation within four weeks of the vaccination.

### 2.3. Survey Implementation

Social media (e.g., Facebook and WhatsApp) was primarily used to recruit the participants. The potential participants were first asked via WhatsApp if they had received at least the first dose of the inactivated COVID-19 vaccine; if the answer was “Yes”, the potential participants were then briefly informed of the study’s aim, and the survey link was sent to them. Moreover, Facebook was used to recruit the participants; the research team posted the survey link using their accounts, and a question about receiving at least the first dose of the inactivated COVID-19 vaccine along with the aim had to be answered “Yes” for them to proceed to the other survey sections. The survey was designed to be completed within an average of five minutes. Eligible participants could view the ethics committee’s approval before filling out the survey.

### 2.4. Sample Size

The sample size calculated in this study was needed to reveal the side effects experienced by people in Jordan after receiving the inactivated COVID-19 vaccine. After considering the number of vaccinated individuals in Jordan (obtained from the Jordanian Ministry of Health website at the time of the study) [22], the sample size was calculated using the Epi Info software, using a margin of error of 5%, a confidence level of 95%, an expected frequency of 50%, and a design effect of 1.0. The minimum representative number of participants was 384 [23].

### 2.5. Statistical Analysis

Following the data collection, the data were analysed using the Statistical Package for the Social Sciences (SPSS), Version 24.0 (IBM Corp., Armonk, NY, USA). Qualitative variables are presented as percentages. A *p*-value of ≤0.05 was deemed to be statistically significant (Chi-square test).

Logistic regression was conducted to screen for the variables (sex (0: male, 1: female), age (0: <30 years old, 1: ≥30 years old), smoking status (0: non-smoker, 1: smoker), and whether the participants have been infected before (0: yes, 1: no)) affecting whether the participants’ experienced side effects after receiving the inactivated COVID-19 vaccine. For a simple logistic regression, a variable with a *p*-value less than 0.25 was deemed to be eligible to enter into the multiple logistic regression to explore the independent variables significantly associated with experiencing side effects after receiving the inactivated COVID-19 vaccine. To ensure the absence of multicollinearity among the independent variables, the variables were chosen after confirming their independence by providing tolerance values that were greater than 0.2 and variance inflation factor values that were less than five. For the multiple logistic regression, a variable that has a *p*-value < 0.05 was considered to be statistically significant.

## 3. Results

The responses of 386 participants (108 males and 278 females) were included in the study analysis. Of the participants (n = 386), 149 of them (38.6%) were under 30 years of age. About sixty percent of the participants (n = 229) were married, around three quarters (75.9%, n = 293) were living in Amman, and 94.8% of them had Jordanian nationality. Regarding the participants’ education, most of the participants (86.3%; n = 334) had a graduate or postgraduate degree. About 65% of them were employed (n = 250), and around one-third of the participants were smokers (n = 124). Table 1 shows the detailed demographic characteristics of the study’s participants (n = 386).

Before receiving the inactivated COVID-19 vaccine, 53.0% of the participants were virus-free (had never been infected with COVID-19). As shown in Figure 1, nearly 24.0% of the participants received the first dose of the inactivated COVID-19 vaccine in June 2021, which was followed by those who received it in May 2021 (21.8%).

Around half of the study’s participants (52.8%; n = 204) reported side effects after receiving the inactivated COVID-19 vaccine. The participants were questioned about 23 different side effects. Fatigue was the most reported side effect (85.1%, n = 174) among the participants (n = 204) after receiving the first dose of the inactivated COVID-19 vaccine. Figure 2 shows the percentages of self-reported side effects among those participants (n = 204).

Multiple logistic regression analysis of factors affecting the experience of side effects after the first dose of the inactivated COVID-19 vaccine among the study participants (n = 386) highlighted that the female sex (OR = 0.382, *p*-value = 0.027) is significantly associated with experiencing side effects (Table 2).

A Chi-square test was performed among the participants who reported side effects after the first dose (n = 204) to assess the association between each side effect and the participants’ sex, age, smoking status, and whether they had previously contracted COVID-19 (Table 3). 

There was a significant association between being female and experiencing chills, muscle or joint pain, anorexia, drowsiness, and hair loss. A significant association was also found between being over 30 years of age and experiencing a cough. Being a smoker was significantly associated with experiencing cough and headache. Furthermore, chills and a sore throat were significantly associated with individuals who had never been infected (Figure 3).

Out of the study participants (n = 386), 84.5% of them (n = 326) received the second dose of the inactivated COVID-19 vaccine. More than half of the participants who received the second dose reported side effects (54.6%, n = 178).

As shown in Table 4, fatigue followed by a sore arm at the site of the injection and discomfort were the most reported side effects following the second dose of the inactivated COVID-19 vaccine (80.9%, 77.0%, and 62.9%, respectively).

Only nine participants (2.4%) reported severe side effects that required hospital admission within four weeks of receiving the inactivated COVID-19 vaccine.

## 4. Discussion

This study was conducted to assess the safety of the inactivated COVID-19 vaccine and to reveal the association between certain side effects and different parameters by collecting data on the short-term side effects after receiving the vaccine. About half of the study participants reported experiencing side effects after the first dose of the inactivated COVID-19 vaccine. More than half of the current study’s participants reported seven out of twenty-three side effects. Fatigue was the most commonly reported side effect after both of the doses. Moreover, certain side effects were significantly associated with females.

The vaccines’ side effects, such as fever, fatigue, muscle pain, and injection site inflammation, are considered to be a typical natural response to injecting foreign irritants, which is managed by the body’s innate immune system. Neutrophils and macrophages release cytokines when they identify foreign vaccine particles. Cytokines are the chemical messenger that causes immunological reactions such as a fever and muscle discomfort. Thus, when a vaccine is injected, the cytokine response is what is anticipated to happen [24], indicating that the body is developing the desired immunity [25].

Multiple studies among different populations were conducted to assess the side effects following the COVID-19 vaccine: many of them found that the inactivated COVID-19 vaccine induced fewer side effects than other types of COVID-19 vaccines did [26,27,28,29,30,31]. In this study, about half of the participants did not experience any side effects after the first dose of the inactivated COVID-19 vaccine. The results from a cross-sectional study in the United Arab Emirates conducted to collect data on the side effects following vaccination revealed that 25% of the participants stated that they had no side effects after the first dose [30]. 

After conducting several studies, mild side effects were reported after receiving the inactivated COVID-19 vaccine [16,27,28,29,32]. Fatigue, a sore arm at the injection site, and discomfort were the most reported side effects among the current study participants after the first dose. This finding is consistent with the side effects of the vaccine reported by the WHO [33], and other conducted studies [26], for example, a systemic review was conducted to assess the side effects associated with the COVID-19 vaccine, and it was found that injection site pain is one of the most commonly reported local side effects, while fatigue is one of the most reported systemic side effects [34]. A cross-sectional study was conducted to assess the side effects and perceptions after receiving COVID-19 vaccines in Jordan, and most of the reported side effects were similar to the present study as they were mild and non-life threatening, such as fatigue, a headache, joint pain, myalgia, and chills [32]. 

The Inactivated COVID-19 vaccine appears to be a safe choice, owing to its self-limiting mild side effects [34].

Some side effects might be reported under various conditions unrelated to the vaccine: these are cultural nuances in which some cultures lean toward in some situations. Perhaps some of the side effects (e.g., hair loss) that were reported in Jordan to the vaccine are more related to culture than they are to medical or pharmacological differences. In this study, out of the participants (n = 204) who reported side effects after the first dose, 19.4% of them suffered from hair loss. This side effect has not been reported in any other studies. Other published studies documented related findings, for example, in Italy: three cases of alopecia areata recurrence were reported following the first COVID-19 vaccine dose [35]. Another two cases were identified in China following the second dose of the COVID-19 vaccine for a 29-year-old man who developed balding patches on his scalp and a 26-year-old woman who complained of diffuse hair loss involving scalp, eyelashes, and eyebrows [36]. However, the vaccine type among these cases was not the inactivated COVID-19 vaccine. Further studies should assess the relationship between hair issues and COVID-19 vaccines. 

In this study, side effects related to the inactivated COVID-19 vaccine were significantly associated with the female sex. This result is consistent with other conducted studies, where the side effects associated with the vaccines were more common among females than they are among males [26,29,30,37,38]. The current finding is unsurprising, as the COVID-19 vaccine was not an exception regarding the previous point: females are more likely to experience side effects after receiving various viral and bacterial vaccines [28,29]. This can be explained by the different hormones and genes between males and females, which lead to different immunological responses [39], or it could be due to differences in how females perceive and report symptoms, particularly via unsupervised internet self-reporting.

A significant association was found between being over 30 years old and experiencing a cough. This is similar to a study by Lounis et al., where older participants developed more side effects [26]. Other studies found opposing results, where it was documented in a cross-sectional study that headaches and fatigue were more common among participants who were ≤49 years old. The difference in the current finding could be due to the different age cutoffs, as the age cutoff for the current study was 30 years old, while the other research chose 49 years old to be the age cutoff [30].

Being a smoker was significantly associated with experiencing a cough and a headache among the current study’s participants. As documented in a systemic review, active smoking negatively affects the body’s humoral responses to COVID-19 vaccines, but the pathophysiologic mechanism for this relation is not fully understood [40]. A cross-sectional study was conducted to inspect the potential COVID-19 vaccines’ side effects among 1,180 participants. It was found that smokers were 3.6 times more likely to experience side effects than non-smoker participants were [41]. 

Dar-Odeh et al. conducted a cross-sectional study to assess the long-term adverse events (LTAE) of three COVID-19 vaccines among healthcare providers (dentists and physicians). Among the different types of vaccine, the inactivated COVID-19 vaccine showed the highest signification association with LTAE. The present study assessed the short-term side effects, and more than half of the participants who reported side effects after the first dose documented experiencing fatigue, muscle and joint pain, a headache, and drowsiness, however, the same previously mentioned side effects were reported as LTAE in the study conducted by Dar-Odeh et al. [42]. (see Appendix A).

This study comes with limitations. Since the current study was based on an online questionnaire; this might be a source of selection bias. Moreover, COVID-19 infection after vaccination was not assessed, hence, there could have been co-incidence between the reported side effects and the symptoms of infection. Additionally, muscle and joint pain have been combined as one side effect, however, future studies may separate them into two different side effects due to inconsistent aetiology. Finally, although the study met the minimum calculated number of participants, future studies can use a larger sample size to generalise the results further.

Knowledge of COVID-19 vaccine safety is crucial to reduce public hesitancy to receive the vaccine [34]. Several studies show a large variability in vaccine acceptance, public awareness, and perception of the COVID-19 vaccine [43,44]. Therefore, providing accurate data regarding the vaccines’ side effects would increase public confidence and awareness, which would pave the way for pandemic control.

## Figures and Tables

**Figure 1 vaccines-11-00044-f001:**
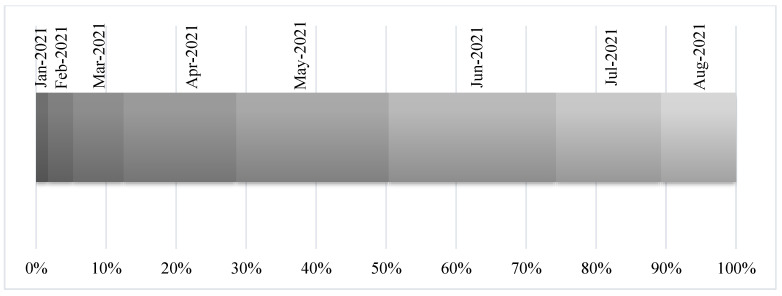
The month in which the inactivated COVID-19 vaccine was received among the study participants (n = 386).

**Figure 2 vaccines-11-00044-f002:**
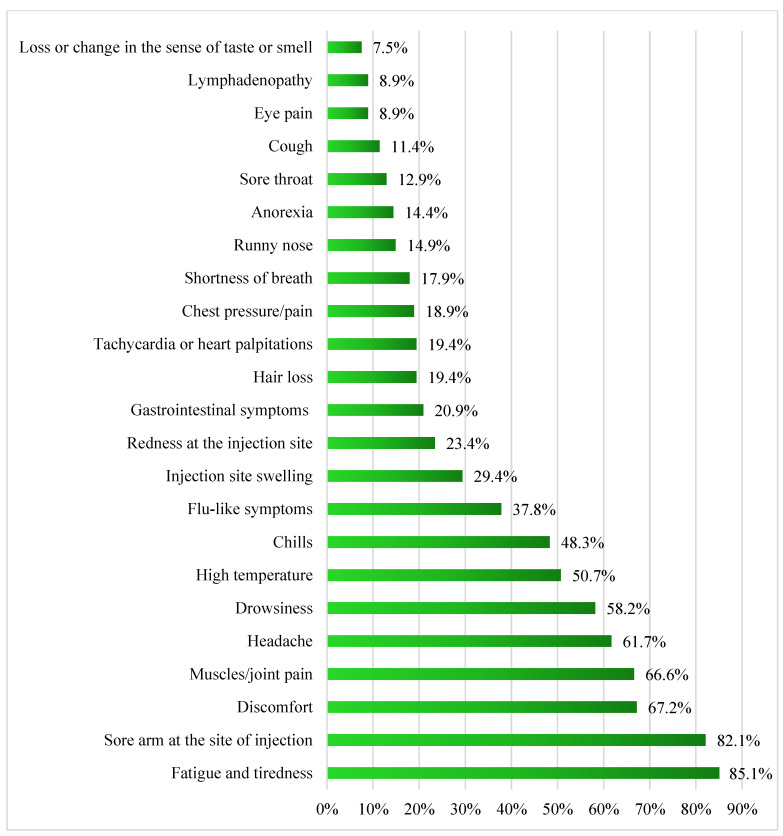
The reported side effects by the participants (n = 204) after the first dose of the inactivated COVID-19 vaccine.

**Figure 3 vaccines-11-00044-f003:**
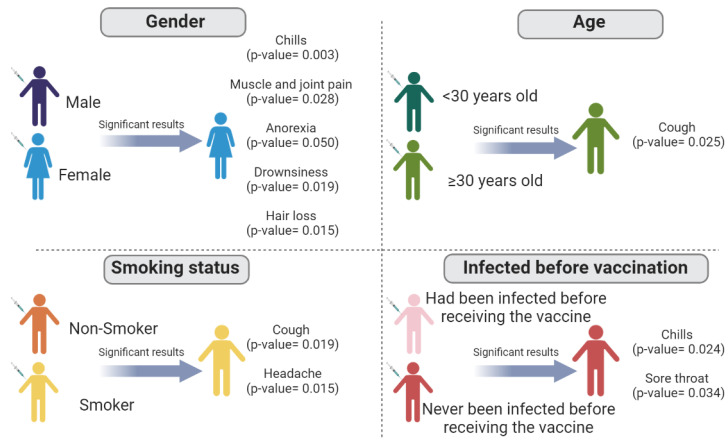
Association between inactivated COVID-19 vaccine side effects and significant demographic factors.

**Table 1 vaccines-11-00044-t001:** Demographic characteristics of the study’s participants (n = 386).

Parameter	n (%)
**Sex** Male Female	108 (28.1) 278 (71.9)
**Age** 18–29 years old 30–39 years old 40–49 years old 50–59 years old ≥60 years	149 (38.6) 155 (40.2) 63 (16.3) 15 (3.9) 4 (1.0)
**Marital Status** Married Single Divorced Widowed	229 (59.3) 141 (36.5) 14 (3.7) 2 (0.5)
**Living place** Amman (the capital) Other cities	293 (75.9) 93 (24.1)
**Nationality** Jordanian Non-Jordanian	366 (94.8) 20 (5.2)
**Educational Level** Primary school High school Diploma Bachelor’s degree Postgraduate degree (Master’s or Ph.D.)	2 (0.5) 22 (5.8) 28 (7.3) 235 (60.6) 99 (25.7)
**Employment** Employed Not employed Retired	250 (64.8) 125 (32.3) 11 (2.9)
**Smoking Status** Smoker Non-smoker Previous smoker	124 (32.0) 250 (64.8) 12 (3.1)

**Table 2 vaccines-11-00044-t002:** Assessment of factors affecting the experience of side effects after the first dose of the inactivated COVID-19 vaccine among study participants (n = 386).

Parameter	Side Effects [0: No, 1: Yes]			
	OR	*p*-Value ^#^	OR	*p*-Value ^$^
Sex Male Female	Reference 0.438	0.003 ^^^	0.382	0.027 *
Age <30 years old ≥30 years old	Reference 0.897	0.606	---	---
Smoking status Non-smoker Smoker	Reference 1.121	0.603	----	----
Infected before vaccination Yes No	Reference 0.697	0.081 ^^^	0.700	0.085

# Using simple logistic regression; ^$^ Using multiple logistic regression; ^ Eligible for entry in multiple logistic regression (significant at 0.25 significance level); * Significant at 0.05 significance level.

**Table 3 vaccines-11-00044-t003:** The reported side effects by the participants (n = 204) after the first dose of the inactivated COVID-19 vaccine.

Reported Side Effects	Sex (0: Male; 1: Female)	Age (0: <30 Years Old; 1: ≥30 Years Old)	Smoking Status (0: Non-Smoker; 1: Smoker)	Infected before the Vaccine (0: Yes; 1: No)
	*p*-Value			
**Sore arm at the site of injection**	0.684	0.801	0.450	0.588
Injection site swelling	0.908	0.870	0.546	0.896
Redness at the injection site	0.514	0.435	0.588	0.898
Discomfort feeling	0.840	0.402	0.275	0.250
Fatigue	0.105	0.704	0.591	0.621
Flu-like symptoms	0.481	0.711	0.889	0.167
High temperature	0.439	0.327	0.290	0.633
Chills	0.003 *	0.643	0.212	0.024 *
Headache	0.536	0.485	0.015 *	0.888
Shortness of breath	0.209	0.801	0.450	0.125
Cough	0.712	0.025 *	0.019 *	0183
Muscles/joint pain	0.028 *	0.610	0.589	0.686
Gastrointestinal symptoms such as nausea, vomiting, and diarrhea	0.166	0.543	0.442	0.605
Sore throat	0.820	0.780	0.656	0.034 *
Eye pain	0.157	0.354	0.439	0.976
Runny nose	0.806	0.216	0.749	0.268
Loss or change in the sense of taste or smell	0.282	0.987	0.828	0.697
Anorexia	0.050 *	0.550	0.372	0.502
Chest pressure/pain	0.545	0.747	0.618	0.392
Drowsiness	0.019 *	0.060	0.367	0.760
Hair loss	0.015 *	0.573	0.081	0.153
Tachycardia or heart palpitations	0.902	0.590	0.055	0.442
Lymphadenopathy	0.157	0.934	0.811	0.976

* Chi-square test (significant at 0.05 significance level).

**Table 4 vaccines-11-00044-t004:** The reported side effects by the participants (n = 178) after the second dose of the inactivated COVID-19 vaccine.

Reported Side Effect	n (%)
Fatigue and tiredness	144 (80.9)
Sore arm at the site of injection	137 (77.0)
Discomfort	112 (62.9)
Muscles/joint pain	105 (59.0)
Headache	99 (55.6)
Drowsiness	90 (50.6)
High temperature	69 (38.8)
Chills	67 (37.6)
Flu-like symptoms	54 (30.3)
Injection site swelling	50 (28.1)
Redness at the injection site	39 (21.9)
Gastrointestinal symptoms such as nausea, vomiting, and diarrhea	34 (19.1)
Hair loss	32 (18.0)
Tachycardia or heart palpitations	30 (16.8)
Chest pressure/pain	29 (16.3)
Shortness of breath	28 (15.7)
Runny nose	23 (12.9)
Anorexia	20 (11.2)
Sore throat	19 (10.8)
Cough	16 (9.0)
Eye pain	15 (8.4)
Lymphadenopathy	13 (7.3)
Loss or change in the sense of taste or smell	10 (5.6)

## Data Availability

Data available on request due to privacy or ethical restrictions.

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
