# Peer review of "Web-Based Reporting of Post-Vaccination Symptoms for Inactivated COVID-19 Vaccines in Jordan: A Cross-Sectional Study"

_vaccines, 2022, doi:10.3390/vaccines11010044_

Round 1
Reviewer 1 Report
Major concerns.
1. Have you asked about COVID-19 infection after vaccination or excluded participants who got COVID-19 infection? Because the outcomes (side effects) may co-incident with infection too.
Minor concerns.
1. Tables. Please align the bullet point and text in the first column to the left side to make it nice and orderly.
2. Figure 1. If it was possible. Could you divide the variable "Muscle/joint pain" into 1) Muscle pain and 2) Joint pain?
Because the aetiology of these outcomes may not be consistent. If it is not, maybe a limitation during data collection.
3. Please use the "p-value" in only one of the formats throughout the manuscript; do not use both. (p or P and with italicised or non-italicised)
The Tables used "P", and Figure 2 used "p".
Comments.
1. Line 143, "(IBM Corp., Armonk, New York, USA)". Suggest using "NY" instead of New York State.
Author Response
I appreciate the suggestions, which have been very helpful in improving the manuscript. I also thank you for your careful reading of the manuscript
All the received comments on this study have been taken into account in improving the quality of the article, and I present below the reply to each of them separately.
Reviewer 1
Major concerns.
C1. Have you asked about COVID-19 infection after vaccination or excluded participants who got COVID-19 infection? Because the outcomes (side effects) may co-incident with infection too.
R1. This was added to the study limitation as follows: “This study comes with limitations, since the current study was based on an online questionnaire, this might be a source of selection bias. Moreover, COVID-19 infection after vaccination was not assessed, hence, there could have been a co-incident between the reported side effects and the symptoms of infection. Additionally, muscle and joint pain have been combined as one side effect; however, future studies may separate them into two different side effects due to inconsistent aetiology.”
Minor concerns.
C1. Tables. Please align the bullet point and text in the first column to the left side to make it nice and orderly.
R1. Done. The text has been aligned as requested.
C2. Figure 1. If it was possible. Could you divide the variable "Muscle/joint pain" into 1) Muscle pain and 2) Joint pain?
Because the aetiology of these outcomes may not be consistent. If it is not, maybe a limitation during data collection.
R2. This point was added to the study limitation as follows: “Additionally, muscle and joint pain have been combined as one side effect; however, future studies may separate them into two different side effects due to inconsistent aetiology”
C3. Please use the "p-value" in only one of the formats throughout the manuscript; do not use both. (p or P and with italicised or non-italicised)
The Tables used "P", and Figure 2 used "p".
R3. Done. The following point has been edited. It was written as follows “p-value” throughout the manuscript.
Comments.
C1. Line 143, "(IBM Corp., Armonk, New York, USA)". Suggest using "NY" instead of New York State.
R1. Done. “New York” was changed to “NY”
Reviewer 2 Report
what kind of inactivated COVID-19 vaccines did patient took?
How the authors calculated the number of respondents they need?
To me the sample size is quite low to reach a final conclusion.
How they validate Survey questions? Authors describe the process in line 113-119. But its not sufficient.
Author Response
I appreciate the suggestions, which have been very helpful in improving the manuscript. I also thank you for your careful reading of the manuscript
All the received comments on this study have been taken into account in improving the quality of the article, and I present below the reply to each of them separately.
Reviewer 2
C1. what kind of inactivated COVID-19 vaccines did patient took?
R1. The following sentence “A cross-sectional study design was carried out in August 2021 to collect data regarding the side effects following the inactivated COVID-19 vaccine, among the Jordanian population” has been changed as follows: “A cross-sectional study was carried out in August 2021 to collect data regarding the side effects of the inactivated COVID-19 vaccine (Sinopharm/Beijing®), among the Jordanian population”
C2. How the authors calculated the number of respondents they need?
R2. The number of vaccinated individuals in Jordan was obtained from the Jordanian Ministry of Health website at the time of the study, the rest of the sample size calculation is clarified.
The sample size calculation was clarified as edited: “After considering the number of vaccinated individuals in Jordan (obtained from the Jordanian Ministry of Health website at the time of the study) (22), the sample size was calculated using the Epi Info software, using a margin of error of 5%, confidence level of 95%, expected frequency of 50%, and a design effect of 1.0. The minimum representative number of participants was 384 (23).”
C3. To me the sample size is quite low to reach a final conclusion.
R3. The following sentence was added to the limitation section: “Finally, although the study met the minimum calculated number of participants, future studies can use a larger sample size to generalise the results further.”
C4. How they validate Survey questions? Authors describe the process in line 113-119. But its not sufficient.
C4. The following paragraph was added to clarify the face and content validity of the survey: “To ensure face and content validity, the first draft of the survey was validated by an expert panel that evaluated questions' comprehension, relevancy, and word clarity. The expert panel included five independent academics randomly selected from a list of 30 academics who worked at different higher education institutions. The academics were selected based on their experience (minimum 10 years) in related research areas. In addition to the five academics, two specialist pulmonologists were requested to validate the survey. The experts were invited to assess the suitability of the words, appropriateness of the content, consistency of the layout and style, relevancy of the survey items to the study objectives, and whether the items are related to the study's aim. Furthermore, they confirmed the survey is free from medical jargon and complicated terminology. Amendments were conducted based on their feedback. A pilot for the survey was conducted, and necessary refinements were made. The survey questions were revised as a final point in the development and to make the study appropriate for online administration.”
Reviewer 3 Report
This paper surveys reported symptoms following COVID vaccination in Jordan. It is important to understand the Jordanian experience with COVID vaccination and Jordanian attitudes towards vaccination. Therefore this paper fills in an important gap in knowledge about COVID in all the countries of the world, assuming that these results add value to existing publications on vaccination symptoms in Jordan.
MAJOR COMMENTS:
Can you comment on how attitudes might have evolved over time in Jordan?
maybe Jordanians were less hesitant early in the pandemic and more hesitant late. Or perhaps the other way around...
_______________
Please frame almost all results as "reported side effects" rather than "side effects". Because you are relying on user's Internet reporting, and do not have an objective professional measuring these, there could be a difference between real and reported side effects. For example, the column header in Table 4 should be "Reported Side Effect" rather than "Side Effect". Check to make sure all mentions of "side effect" in the paper are changed to "reported side effect" if that is your meaning.
In particular, you will want to replace the word 'experiencing' with 'reporting' throughout the Abstract and Manuscript.
______
Are there any side effects that are typical side effects in Jordan? i.e., side effects that might be reported by folks under a variety of conditions - not necessarily related to vaccination? For example, it is common for Russian men to complain of chest pain and for Chinese women to complain of stomach pain. These are cultural nuances that folks in these cultures lean towards in some situations - for example when they are depressed. The side effects reported in Jordan to the vaccine might be more related to culture than any medical/pharmacological differences. You should at least acknowledge this possibility even if you have no data related to culturally specific side effects. For example, I find the female reporting hair loss data very surprising. I wonder if it is cultural.
_________
19% reported hair loss. That seems huge. Can you find any other articles that describe such a huge rate of hair loss, or is has this only been seen in Jordan? If only in Jordan, you will want to explain why you think it is so. Do you think perhaps Jordanian women who are internet connected are very focused on their hair?
________________________________________________
It is unclear why you did not cite these two articles. Please include them and compare and contrast your results to their results.
I did a PubMed search
jordan [titl] AND COVID AND vaccination AND symptom
and found 7 results, including these two articles.
It was so easy to find them, I worry that you have not systemically scoured the litterature to find other references that might also be relevant:
Dar-Odeh N, Abu-Hammad O, Qasem F, Jambi S, Alhodhodi A, Othman A, Abu-Hammad A, Al-Shorman H, Ryalat S, Abu-Hammad S. Long-term adverse events of three COVID-19 vaccines as reported by vaccinated physicians and dentists, a study from Jordan and Saudi Arabia. Hum Vaccin Immunother. 2022 Dec 31;18(1):2039017. doi: 10.1080/21645515.2022.2039017. Epub 2022 Mar 3. PMID: 35240939; PMCID: PMC9009903.
Hatmal MM, Al-Hatamleh MAI, Olaimat AN, Hatmal M, Alhaj-Qasem DM, Olaimat TM, Mohamud R. Side Effects and Perceptions Following COVID-19 Vaccination in Jordan: A Randomized, Cross-Sectional Study Implementing Machine Learning for Predicting Severity of Side Effects. Vaccines (Basel). 2021 May 26;9(6):556. doi: 10.3390/vaccines9060556. PMID: 34073382; PMCID: PMC8229440.
__________________________________
MINOR POINTS
TITLE
"Assessment of the safety of the inactivated COVID-19 vaccines, and the association between experiencing side effects and different parameters.
This article is not really about the safety of vaccines. And phrases like "different parameters" are so vague as to be worthless in a title.
Change to:
"Web-based reporting of post-vaccination symptoms in Jordan for inactivated COVID-19 vaccines: a cross-sectional study"
__________
Abstract
"The rapid development of COVID-19 vaccines has been identified as a major barrier to receiving the vaccine."
No. The opposite is true. If the vaccine had NOT been developed - THAT would have been a major barrier. I think you mean to say something like "Perception of vaccines as being unsafe is a major barrier to receiving the vaccine."
_________
typo:
"Proving the public with accurate data regarding vaccines reduces vaccine hesitancy."
->
"Providing the public with accurate data regarding vaccines reduces vaccine hesitancy."
___
"Data were analyzed using the Statistical Package for Social Science (SPSS)."
delete this sentence. It is trivia that does not belong in the Abstract. You can put it in Methods if you wish.
________
INTRODUCTION
""The technology behind developing inactivated vaccines is very traditional and has led to numerous vaccines in the past including seasonal influenza vaccines (2)."
Traditional means something different than what you seem to think it means. So just delete that comment.
->
"The technology behind developing inactivated vaccines has led to numerous vaccines in the past including seasonal influenza vaccines (2)."
___________
"This is the first study conducted among the Jordanian population to assess the side effects specifically associated with the inactivated COVID-19 vaccine. Other published studies conducted in Jordan assessed different types of COVID-19 vaccines with none focusing only on the inactivated COVID-19 vaccine."
But you still need to compare your results with these other studies. And to cite them.
METHODS
how did you advertise your study? word of mouth? paid advertisements on the web? printed flyers? etc.
I see that you write "Social media (e.g., Facebook and WhatsApp) was primarily used to recruit the participants."
But how?
For example, did you pick specific Facebook groups to post messages in and/or did you do a targeted advertisement buy with Facebook? If you picked specific groups, which groups? Same questions for WhatsApp.
___
was informed consent in person or remote? Can you provide a copy of the blank informed consent as supplemental material? And also the questionnaire as supplemental material?
___
"was validated by an expert panel"
describe this expert panel. How were they selected? Are they named in the Acknowledgements?
____
"The sample size was calculated after considering the number of vaccinated individuals in Jordan, using a"
The sample size for what? Testing a hypothesis? What hypothesis?
_____
RESULTS
give more details on when the survey was conducted (span of months) and the full breakdown of when people in the survey were vaccinated. Perhaps a bar graph.
___
FIGURE 2 Legend
"Figure 2. Association between inactivated COVDI-19 vaccine side effects and different parameters."
'parameters' is a really vague word. How about (and correct typo):
"Figure 2. Association between inactivated COVID-19 vaccine side effects and significant demographic factors."
_____________
DISCUSSION
____________
"Differentiating myth from truth, no available vaccine is believed to be 100% side effects-free (18)."
Delete this sentence. It is obvious and does not add anything to the paper.
________________________
"Vaccines’ side effects such as fever, fatigue, muscle pain, and injection site inflammation are considered a common natural response to the injection of foreign drugs,"
A vaccine is not really considered a drug. Perhaps you mean
"Vaccines’ side effects such as fever, fatigue, muscle pain, and injection site inflammation are considered a common natural response to the injection of foreign irritants,"
___
"This can be explained by the different hormones and genes between males and females which lead to different immunological responses (32)."
Change this to
"This can be explained by the different hormones and genes between males and females which lead to different immunological responses (32), or could be due to differences in how females perceive and report symptoms, particularly via unsupervised internet self-reporting."
____
"Furthermore, it was stated by the authors that the inactivated vaccine appears to be a safe choice owing to its self-limiting mild side effects (29)."
wordy, just write:
"The inactivated vaccine appears to be a safe choice owing to its self-limiting mild side effects (29)."
___
"As documented in a systemic review, active smoking negatively affects the body’s humoral responses to COVID-19 vaccines, however, the pathophysiologic mechanism for this relation has not been entirely proposed (33)."
wordy & awkward, rephrase as:
"As documented in a systemic review, active smoking negatively affects the body’s humoral responses to COVID-19 vaccines, but the pathophysiologic mechanism for this relation is not fully understood (33)."
____
I would expand your "limitations paragraph and move it earlier. End your paper with the stronger paragrpah that currently precedes it. Online reporting is not just a limitation because of selection bias, but also can change how reliably and truthfully people recall and report their symptoms.
___
Author Response
I appreciate the suggestions, which have been very helpful in improving the manuscript. I also thank you for your careful reading of the manuscript
All the received comments on this study have been taken into account in improving the quality of the article, and I present below the reply to each of them separately.
Reviewer 3
This paper surveys reported symptoms following COVID vaccination in Jordan. It is important to understand the Jordanian experience with COVID vaccination and Jordanian attitudes towards vaccination. Therefore, this paper fills in an important gap in knowledge about COVID in all the countries of the world, assuming that these results add value to existing publications on vaccination symptoms in Jordan.
MAJOR COMMENTS:
C1. Can you comment on how attitudes might have evolved over time in Jordan?
maybe Jordanians were less hesitant early in the pandemic and more hesitant late. Or perhaps the other way around...
R1. Done. The following paragraph was added to the Introduction:
“People's attitudes towards vaccination have changed over the era of the COVID-19 pandemic. No specific study has looked closely into this change of attitude; however, an indirect evaluation of reported beliefs can provide such insight. A cross-sectional study (August 2020) was conducted to evaluate the perception of people in Jordan regarding the COVID-19 vaccines and assess their hesitancy toward receiving the COVID-19 vaccine. Of the participants (n= 1287), more than half (n= 665) reported not having adequate information about the COVID-19 vaccine benefits. Moreover, 64% preferred to achieve natural immunity (18). In December 2020, a cross-sectional study was conducted to assess attitudes toward COVID-19 vaccines among several countries, and the acceptance rate for the COVID-19 vaccine in Jordan was found to be 28.4% (19). One explanation for the low COVID-19 vaccine acceptance could be conspiracy beliefs which appear to worsen over time. A cross-sectional study conducted in April 2020 revealed that 47.9% of the participants (n= 3150) think the COVID-19 pandemic is a part of a global conspiracy theory (20). After eight months, in December 2020, another cross-sectional study was conducted, and the percentage increased, as 58.5% of participants believed that COVID-19 is a -man-made disease (19). Such discrepancy in COVID-19 vaccine acceptance might be attributed to the participants' age, gender, background, and educational level; for example, the study findings of a cross-sectional study conducted to assess attitudes and perceptions of Jordanian healthcare providers to the COVID-19 vaccine showed that acceptance level ranges from 42.6% for nurses to 83.3% for physicians (21).”
_______________
C2. Please frame almost all results as "reported side effects" rather than "side effects". Because you are relying on user's Internet reporting, and do not have an objective professional measuring these, there could be a difference between real and reported side effects. For example, the column header in Table 4 should be "Reported Side Effect" rather than "Side Effect". Check to make sure all mentions of "side effect" in the paper are changed to "reported side effect" if that is your meaning.
In particular, you will want to replace the word 'experiencing' with 'reporting' throughout the Abstract and Manuscript.
R2. Done. The following point was edited throughout the manuscript as suggested.
______
C3. Are there any side effects that are typical side effects in Jordan? i.e., side effects that might be reported by folks under a variety of conditions - not necessarily related to vaccination? For example, it is common for Russian men to complain of chest pain and for Chinese women to complain of stomach pain. These are cultural nuances that folks in these cultures lean towards in some situations - for example when they are depressed. The side effects reported in Jordan to the vaccine might be more related to culture than any medical/pharmacological differences. You should at least acknowledge this possibility even if you have no data related to culturally specific side effects. For example, I find the female reporting hair loss data very surprising. I wonder if it is cultural.
R3. Thank you for bringing our attention to this idea.
Done. The following paragraph was added: “Some side effects might be reported under various conditions unrelated to the vaccine; these are cultural nuances that some cultures lean toward in some situations. Perhaps some of the side effects (e.g., hair loss) that were reported in Jordan to the vaccine are more related to culture than medical or pharmacological differences. In this study, out of the participants (n= 204) who reported side effects after the first dose, 19.4% suffered from hair loss. This side effect has not been reported in any other studies. Other published studies documented related findings; for example, in Italy, three cases of alopecia areata recurrence were reported following the first COVID-19 vaccine dose (36). Another two cases were identified in China following the second dose of the COVID-19 vaccine for a 29-year-old man who developed balding patches on the scalp and a 26-year-old woman who complained of diffuse hair loss involving scale, eyelashes, and eyebrows (37). However, the vaccine type among these cases was not the inactivated COVID-19 vaccine. Further studies should assess the relationship between hair issues and COVID-19 vaccines.”
_________
C4. 19% reported hair loss. That seems huge. Can you find any other articles that describe such a huge rate of hair loss, or is has this only been seen in Jordan? If only in Jordan, you will want to explain why you think it is so. Do you think perhaps Jordanian women who are internet connected are very focused on their hair?
R4. The following paragraph was added: “Some side effects might be reported under various conditions unrelated to the vaccine; these are cultural nuances that some cultures lean toward in some situations. Perhaps some of the side effects (e.g., hair loss) that were reported in Jordan to the vaccine are more related to culture than medical or pharmacological differences. In this study, out of the participants (n= 204) who reported side effects after the first dose, 19.4% suffered from hair loss. This side effect has not been reported in any other studies. Other published studies documented related findings; for example, in Italy, three cases of alopecia areata recurrence were reported following the first COVID-19 vaccine dose (36). Another two cases were identified in China following the second dose of the COVID-19 vaccine for a 29-year-old man who developed balding patches on the scalp and a 26-year-old woman who complained of diffuse hair loss involving scale, eyelashes, and eyebrows (37). However, the vaccine type among these cases was not the inactivated COVID-19 vaccine. Further studies should assess the relationship between hair issues and COVID-19 vaccines.”
________________________________________________
C5. It is unclear why you did not cite these two articles. Please include them and compare and contrast your results to their results.
I did a PubMed search
jordan [titl] AND COVID AND vaccination AND symptom
and found 7 results, including these two articles.
It was so easy to find them, I worry that you have not systemically scoured the litterature to find other references that might also be relevant:
Dar-Odeh N, Abu-Hammad O, Qasem F, Jambi S, Alhodhodi A, Othman A, Abu-Hammad A, Al-Shorman H, Ryalat S, Abu-Hammad S. Long-term adverse events of three COVID-19 vaccines as reported by vaccinated physicians and dentists, a study from Jordan and Saudi Arabia. Hum Vaccin Immunother. 2022 Dec 31;18(1):2039017. doi: 10.1080/21645515.2022.2039017. Epub 2022 Mar 3. PMID: 35240939; PMCID: PMC9009903.
Hatmal MM, Al-Hatamleh MAI, Olaimat AN, Hatmal M, Alhaj-Qasem DM, Olaimat TM, Mohamud R. Side Effects and Perceptions Following COVID-19 Vaccination in Jordan: A Randomized, Cross-Sectional Study Implementing Machine Learning for Predicting Severity of Side Effects. Vaccines (Basel). 2021 May 26;9(6):556. doi: 10.3390/vaccines9060556. PMID: 34073382; PMCID: PMC8229440.
R5. Thank you for suggesting these two references.
The suggested references were added to the manuscript.
And the following paragraphs were added to compare and contrast the results:
“Dar-Odeh et al. conducted a cross-sectional study to assess long-term adverse events (LTAE) of three COVID-19 vaccines among healthcare providers (dentists and physicians). Among the different types of vaccine, the inactivated COVID-19 vaccine should the highest signification association with the LTAE. The present study assessed the short-term side effects, and more than half of the participants who reported side effects after the first dose documented experiencing fatigue, muscle and joint pain, headache, and drowsiness; however, the same previously mentioned side effects were reported as LTAE in the study conducted by Dar-Odeh et al. (42).”
“A cross-sectional study was conducted to assess the side effects and perceptions after receiving COVID-19 vaccines in Jordan, and most of the reported side effects were similar to the present study as they were mild and non-life threatening such as fatigue, headache, joint pain, myalgia, and chills (32).”
__________________________________
MINOR POINTS
TITLE
C6. "Assessment of the safety of the inactivated COVID-19 vaccines, and the association between experiencing side effects and different parameters.
This article is not really about the safety of vaccines. And phrases like "different parameters" are so vague as to be worthless in a title.
Change to:
"Web-based reporting of post-vaccination symptoms in Jordan for inactivated COVID-19 vaccines: a cross-sectional study"
R6. Done, the title was changed as follows: “Web-based reporting of post-vaccination symptoms for inactivated COVID-19 vaccines in Jordan: a cross-sectional study”
__________
Abstract
C7. "The rapid development of COVID-19 vaccines has been identified as a major barrier to receiving the vaccine."
No. The opposite is true. If the vaccine had NOT been developed - THAT would have been a major barrier. I think you mean to say something like "Perception of vaccines as being unsafe is a major barrier to receiving the vaccine."
R7. The following sentence: “The rapid development of COVID-19 vaccines has been identified as a major barrier to receiving the vaccine”
was changed to “Perception of COVID-19 vaccines as being unsafe is a major barrier to receiving the vaccine.”
_________
typo:
C8. "Proving the public with accurate data regarding vaccines reduces vaccine hesitancy."
->
"Providing the public with accurate data regarding vaccines reduces vaccine hesitancy."
R8. Thank you for bringing our attention to this typo, it was corrected.
___
C9. "Data were analyzed using the Statistical Package for Social Science (SPSS)."
delete this sentence. It is trivia that does not belong in the Abstract. You can put it in Methods if you wish.
R9. Done. The sentence was deleted from the abstract.
_______
INTRODUCTION
C10. "The technology behind developing inactivated vaccines is very traditional and has led to numerous vaccines in the past including seasonal influenza vaccines (2)."
Traditional means something different than what you seem to think it means. So just delete that comment.
->
"The technology behind developing inactivated vaccines has led to numerous vaccines in the past including seasonal influenza vaccines (2)."
R11. Done. The sentence was edited as suggested.
___________
C11. "This is the first study conducted among the Jordanian population to assess the side effects specifically associated with the inactivated COVID-19 vaccine. Other published studies conducted in Jordan assessed different types of COVID-19 vaccines with none focusing only on the inactivated COVID-19 vaccine."
But you still need to compare your results with these other studies. And to cite them.
R11. Done. Other studies have been added to the discussion part, and the different results were compared.
___________
METHODS
C12. how did you advertise your study? word of mouth? paid advertisements on the web? printed flyers? etc.
I see that you write "Social media (e.g., Facebook and WhatsApp) was primarily used to recruit the participants."
But how?
For example, did you pick specific Facebook groups to post messages in and/or did you do a targeted advertisement buy with Facebook? If you picked specific groups, which groups? Same questions for WhatsApp.
R12. The following paragraph was added: “Social media (e.g., Facebook and WhatsApp) was primarily used to recruit the participants. Potential participants were first asked via WhatsApp if they had received at least the first dose of the inactivated COVID-19 vaccine, if the answer was “Yes”, the potential participants were then briefly informed of the study’s aim, and the survey link was sent to them. Moreover, Facebook was used to recruit participants; the research team posted in their account the survey link, and a question about receiving at least the first dose of the inactivated COVID-19 vaccine along with the aim had to be answered "Yes" to proceed to the other survey sections.”
___
C13. was informed consent in person or remote? Can you provide a copy of the blank informed consent as supplemental material? And also the questionnaire as supplemental material?
R13. Done. The informed consent and the questionnaire were added to the supplemental material.
___
C14. "was validated by an expert panel"
describe this expert panel. How were they selected? Are they named in the Acknowledgements?
R14. The following sentence was added: “To ensure face and content validity, the first draft of the survey was validated by an expert panel that evaluated questions' comprehension, relevancy, and word clarity. The expert panel included five independent academics randomly selected from a list of 30 academics who worked at different higher education institutions. The academics were selected based on their experience (minimum 10 years) in related research areas. In addition to the five academics, two specialist pulmonologists were requested to validate the survey.”
Acknowledgement was edited as follows:
“The research team acknowledge the expert academics and pulmonologists for their efforts in validating the survey”
C15. "The sample size was calculated after considering the number of vaccinated individuals in Jordan, using a"
The sample size for what? Testing a hypothesis? What hypothesis?
R15. The following sentence was added to the sample size:
“The sample size calculated in this study is needed to reveal the side effects experienced by people in Jordan after receiving the inactivated COVID-19 vaccine.”
RESULTS
C16. give more details on when the survey was conducted (span of months) and the full breakdown of when people in the survey were vaccinated. Perhaps a bar graph.
R16. Figure one was added which shows the percentage of participants that were vaccinated in each month (from January to August 2021)
___
FIGURE 2 Legend
C17. "Figure 2. Association between inactivated COVDI-19 vaccine side effects and different parameters."
'parameters' is a really vague word. How about (and correct typo):
"Figure 2. Association between inactivated COVID-19 vaccine side effects and significant demographic factors."
R17. Done. The legend was edited as suggested.
_____________
DISCUSSION
____________
C18. "Differentiating myth from truth, no available vaccine is believed to be 100% side effects-free (18)."
Delete this sentence. It is obvious and does not add anything to the paper.
R18. Done. The sentence was deleted.
________________________
C19. "Vaccines’ side effects such as fever, fatigue, muscle pain, and injection site inflammation are considered a common natural response to the injection of foreign drugs,"
A vaccine is not really considered a drug. Perhaps you mean
"Vaccines’ side effects such as fever, fatigue, muscle pain, and injection site inflammation are considered a common natural response to the injection of foreign irritants,"
R19. The sentence was edited as suggested, as follows: “Vaccines’ side effects such as fever, fatigue, muscle pain, and injection site inflammation are considered a typical natural response to injecting foreign irritants”
___
C20. "This can be explained by the different hormones and genes between males and females which lead to different immunological responses (32)."
Change this to
"This can be explained by the different hormones and genes between males and females which lead to different immunological responses (32), or could be due to differences in how females perceive and report symptoms, particularly via unsupervised internet self-reporting."
R20. Done. The sentence was edited as suggested.
____
C21. "Furthermore, it was stated by the authors that the inactivated vaccine appears to be a safe choice owing to its self-limiting mild side effects (29)."
wordy, just write:
"The inactivated vaccine appears to be a safe choice owing to its self-limiting mild side effects (29)."
R21. Done. The sentence was changed.
___
C22. "As documented in a systemic review, active smoking negatively affects the body’s humoral responses to COVID-19 vaccines, however, the pathophysiologic mechanism for this relation has not been entirely proposed (33)."
wordy & awkward, rephrase as:
"As documented in a systemic review, active smoking negatively affects the body’s humoral responses to COVID-19 vaccines, but the pathophysiologic mechanism for this relation is not fully understood (33)."
R22. Done. The sentence was changed.
____
C23. I would expand your "limitations paragraph and move it earlier. End your paper with the stronger paragraph that currently precedes it. Online reporting is not just a limitation because of selection bias, but also can change how reliably and truthfully people recall and report their symptoms.
R23. Done.
___
Round 2
Reviewer 2 Report
Manuscript is much more improved now
Reviewer 3 Report
I think this is OK to publish. I feel the English is still awkward in places. You might want to have an English expert polish your manuscript.